# Point-of-Care Ultrasound (POCUS) as an Extension of the Physical Examination in Patients with Bacteremia or Candidemia

**DOI:** 10.3390/jcm11133636

**Published:** 2022-06-23

**Authors:** Serafín López Palmero, Miguel Angel López Zúñiga, Virginia Rodríguez Martínez, Raul Reyes Parrilla, Ana Maria Alguacil Muñoz, Waldo Sánchez-Yebra Romera, Patricia Martín Rico, Inmaculada Poquet Catalá, Carlos Jiménez Guardiola, Alfonso Del Pozo Pérez, Ruben Lobato Cano, Ana Maria Lazo Torres, Gines López Martínez, Luis Felipe Díez García, Tesifon Parrón Carreño

**Affiliations:** 1Internal Medicine Department, Torrecárdenas University Hospital, 04009 Almería, Spain; viroma_r3@hotmail.com (V.R.M.); gilomart@gmail.com (G.L.M.); luisf.diez.sspa@juntadeandalucia.es (L.F.D.G.); 2Infectious Diseases Department, Virgen de las Nieves University Hospital, 18014 Granada, Spain; malzuniga@ugr.es; 3Cardiology Department, Torrecárdenas University Hospital, 04009 Almería, Spain; raul.repa@gmail.com; 4Internal Medicine Department, Toledo University Hospital, 45600 Toledo, Spain; amana12i@gmail.com; 5Microbiology Department, Torrecárdenas University Hospital, 04009 Almería, Spain; waldoe.sanchez.sspa@juntadeandalucia.es; 6Internal Medicine Department, Marina Salud Dénia Hospital, 03700 Alicante, Spain; pmrico@yahoo.es (P.M.R.); inmaculada.poquet@marinasalud.es (I.P.C.); 7Internal Medicine Department, Vega Baja Orihuela Hospital, 03314 Alicante, Spain; carlosjimenezguardiola@gmail.com (C.J.G.); alfonsodelpozo@gmail.com (A.D.P.P.); 8Internal Medicine Department, Virgen de las Nieves University Hospital, 18014 Granada, Spain; ruben.lobato27@gmail.com; 9Infectious Diseases Department, Torrecárdenas University Hospital, 04009 Almería, Spain; anamlazotorres@gmail.com; 10Department of Medicine, School of Medicine, Almería University, 04120 Almería, Spain; tesifonparron54@gmail.com

**Keywords:** point-of-care ultrasound (POCUS), bacteremia, candidemia, infective endocarditis

## Abstract

Background: In general, transthoracic echocardiography (TTE) is the first diagnostic test used for patients with bacteremia or candidemia and clinical signs of Infective Endocarditis (IE). Point-of-care ultrasound (POCUS) may be used in addition to physical examination for the detection of structural heart disease and valve abnormalities. Objective: To determine the diagnostic accuracy of POCUS for the detection of signs suggestive of IE, including vegetation, valvular regurgitation, structural heart disease, hepatomegaly, splenomegaly and septic embolisms, in patients with bacteremia or candidemia. Design: Observational, cross-sectional, multicenter study using convenience sampling. Setting: Six Spanish academic hospitals. Patients: Adult patients with bacteremia or candidemia between 1 February 2018 and 31 December 2020. Measurements: The reference test, to evaluate vegetation, valvular regurgitation and structural heart disease, was transesophageal echocardiography (TEE). For patients who did not undergo TEE, transthoracic echocardiography (TTE) was considered the reference test. POCUS was performed by internists, while conventional echocardiography procedures were performed by cardiologists. Results: In 258 patients, for the detection of valvular vegetation, POCUS had sensitivity, specificity, and positive and negative predictive values of 77%, 94%, 82% and 92%, respectively. For valvular regurgitation (more than mild), sensitivity was ≥76% and specificity ≥85%. Sensitivity values for the detection of hepatomegaly and splenomegaly were 92% and 92%, respectively, while those for specificity were 96% and 98%. Conclusion: POCUS could be a valuable tool, as a complement to physical examination, at the hospital bedside for patients with bacteremia or candidemia, helping to identify signs suggestive of IE.

## 1. Introduction

Infective endocarditis (IE) is rare but presents high mortality (18–46%) [1,2]. It is usually diagnosed by the modified Duke criteria, incorporating clinical, microbiological and echocardiographic findings [1,2,3]. The clinical presentation is usually nonspecific, which may delay diagnosis and the initiation of appropriate antimicrobial treatment, leading to increased morbidity and mortality [4].

In patients with candidemia or bacteremia due to microorganisms typical of IE, echocardiography should be performed within 5–7 days from the onset of these conditions [5,6,7]. In general, transthoracic echocardiography (TTE) is the first diagnostic test used for patients with clinical signs of IE. This test has a sensitivity of 45–89% and a specificity >90% [7,8]. The absence of vegetation does not exclude the diagnosis of IE, while its probability is substantially reduced when a valve with normal morphology and function is observed. Therefore, TTE could be sufficient in cases of uncomplicated bacteremia, with a low clinical probability of IE, defined by the following: nosocomial-acquired bacteremia, absence of prosthetic material (e.g., mechanical valve, cardiac electrostimulation), non-dependence on hemodialysis, sterile blood culture samples (2–4 days after starting antimicrobial treatment), defervescence at 72 h after initiating antimicrobial treatment, and the absence of metastatic infection (e.g., brain, abdominal or osteoarticular) [7,8,9,10,11].

Transesophageal echocardiography (TEE) has a sensitivity >90% and can detect cardiac complications such as abscesses, valve perforation and pseudoaneurysms [7,8]. TEE is required in the following clinical scenarios: the presence of a prosthetic valve or a pacemaker, TTE negative or technically inadequate with a strong clinical suspicion of IE, positive TTE and suspicion of intracardiac complications (such as an abscess), and prior to cardiac surgery [4]. However, TEE is associated with an increase in costs, patient discomfort, hospitalization prolongation and lack of prompt availability.

Point-of-care ultrasound (POCUS) is the limited use of ultrasound examination by the physician responsible in order to enhance conventional cardiac physical examination in response to specific clinical issues [12,13,14,15,16,17,18]. POCUS could be useful for detecting echocardiographic signs suggestive of IE (e.g., vegetation, valvular regurgitation or septic embolisms) in patients with bacteremia or candidemia [19,20]. However, there is currently no evidence regarding the value of POCUS as a test for detecting IE in patients with bacteremia or candidemia and the clinical suspicion of IE.

The main aim of the present study was to determine the usefulness of multi-organ POCUS, including cardiac, lung and abdominal ultrasound, as an extension of the physical examination for detecting signs suggestive of IE, including vegetation, valvular regurgitation, and structural heart disease, in patients with bacteremia or candidemia. In addition, we assess the usefulness of POCUS for the detection of hepatomegaly, splenomegaly and septic embolisms.

## 2. Materials and Methods

We conducted an observational, cross-sectional, multicenter study to validate diagnostic tests. The following University Hospitals took part in the study: Torrecardenas Almeria, Jaen, Toledo, Marina Salud Denia, Vega Baja Orihuela and Virgen de las Nieves Granada. The study is in accordance with the Declaration of Helsinki and was approved by the Torrecardenas University Hospital Research Ethics Committee. Informed consent was obtained from each enrolled patient.

### 2.1. Patient Selection

The patients included in the study were all aged ≥18 years, had been admitted to one of the participating hospitals and had provided a positive blood culture (Table 1). Patients in whom echocardiography (TEE or TTE) could not be performed (*n* = 9), who had a bad ultrasound window (*n* = 11) or who did not give informed consent, were excluded from the study.

### 2.2. Sample

The POCUS test has only recently been implemented in the scope of IE, and therefore this was an exploratory study, no prespecified sample size calculation was performed, and we considered a pilot sample of 258 patients.

### 2.3. Epidemiological, Clinical, Laboratory and Ultrasound Data Collection

In each of the participating hospitals, the Microbiology and Internal Medicine departments were coordinated to ensure optimal daily communication of positive blood cultures. Each patient included in the study was identified, assigned a study number and completed a standard data collection form (Appendix B). From the fifth day of the positive blood culture, an internist trained in basic echocardiography (level I according to the recommendations of the Spanish Cardiology Society) performed a clinical assessment (clinical history and physical examination), followed by an examination using POCUS, at the patient’s hospital bedside. The internists participating in the study were experts only in POCUS; they do not perform complete TTE in their routine activity. One internist at each hospital performed all the POCUS studies.

The POCUS examination was performed with the patient lying in the left lateral decubitus position. The images used to assess vegetation, detect significant valvopathies, determine the dimension of cavities and estimate the left ventricle (LV) systolic function were obtained through the parasternal (long and short axes), apical and subcostal planes.

Subsequently, in a period of 24–48 h, a cardiologist expert in echocardiography performed a complete TTE in the echocardiography laboratory, under standard conditions of ambient light and patient position, assisted by auxiliary personnel. We considered a follow-up strategy in patients in which TEE was not performed, with TTE repetition at 7 days, as suggested by guidelines. Follow-up TTE was performed more frequently in patients with persistent bacteremia or candidemia (positive blood cultures 48 h after starting appropriate antimicrobial treatment), and specifically, when blood cultures were positive for *Enterococcus* spp., *Staphylococcus aureus* or *Streptococcus* spp. The TEE was performed in the following circumstances [4]: negative or technically inadequate TTE with strong clinical suspicion of IE, positive TTE, suspected intracardiac complications and prior to cardiac surgery. To ensure independence of the measurements between the two observers, the results were introduced into the database by an internist and cardiologist without knowing the result of the other (i.e., the observers were blinded to each other).

An abdominal imaging test, either ultrasound or computed tomography scan (CT), was applied to 186 patients by a radiodiagnosis specialist to evaluate the presence of hepatomegaly, splenomegaly and images suggestive of septic embolisms in the liver, spleen or kidneys.

The POCUS examination was performed using the Vscan Extend dual ultrasound system (General Electric). Its dimensions are 168 × 76 × 22 mm and its weight is 321 g. It has a sectorial and linear probe, B-mode and Doppler Color. In this study, sectorial probe was used. It has broad bandwidth phased array (1.7–3.8 MHz), field of view for black and white imaging (up to 70 degrees with maximum depth of 24 cm) and color flow sector representing blood flow within an angle of up 40 degrees. With the greater penetration of lower ultrasound frequencies, high-quality harmonic imaging at greater depth can be performed. All images were recorded on a micro-SD memory card and transferred to the computer for review and more complex measurements (Figure 1).

### 2.4. Outcome Measures and Definitions

The dependent variable in our analysis was the diagnosis of IE, which was established using the modified Duke criteria according to the following categories: definite, possible and rejected. Definite IE was diagnosed upon observing the presence of pathological criteria (intracardiac vegetation or abscess with active endocarditis in histology or the identification of microorganisms by culture or histology of intracardiac vegetations or abscesses) or clinical criteria (two major criteria, one major and three minor criteria, or five minor criteria).

The major criteria considered were a positive blood culture (Table 1) and echocardiographic findings (vegetation, abscess, new valvular regurgitation or new prosthetic valve dehiscence).

Minor criteria were predisposing cardiac factors (prior IE, presence of a prosthetic valve or pacemaker, history of heart valve disease or congenital heart disease), non-cardiac predisposing factors (intravenous drug user, vascular catheter, immunosuppression, dental procedure, digestive system endoscopy or recent surgery), fever (temperature ≥38 °C), vascular phenomena (major arterial embolism, septic pulmonary infarction, mycotic aneurysm, intracranial hemorrhage, conjunctival hemorrhage, Janeway lesions), immunological phenomena (glomerulonephritis, Osler nodes, Roth spots, positive rheumatoid factor) and microbiological findings (positive blood culture or serology for microorganisms not included in Table 1).

IE related to health care was considered present in the following circumstances [3]: the appearance of symptoms at 48 h after hospital admission; hospital care in the six months preceding the diagnosis (e.g., intravenous treatment, hemodialysis, hospitalization for more than 48 h, wound care or other specialized care) and long-term stay in a social health center.

Early-onset prosthetic IE was defined as that occurring in the 12 months following the surgery, and late prosthetic IE as that performed at a later date [3].

The independent variables (age, sex, hospital stay, clinical manifestations, laboratory data and echocardiography) are described in Appendix B. The ultrasound variables described are common to POCUS and conventional echocardiography (TTE, TEE).

Vegetation was defined as an irregular echogenic intracardiac mass, oscillating, located within a valve, in implanted material (prosthesis, catheter) or in the regurgitant jet, in the absence of an alternative anatomical explanation. Valvular vegetation was classed as either present or absent. Valve pathology was classified as mild, moderate or severe, according to visual assessment of the grey-scale and Doppler color images obtained (calcification, limited opening, regurgitant jet). For analysis, valvular regurgitation was considered “more than mild valvular regurgitation” and was classed as either present or absent. LV systolic function was assessed qualitatively, by visual estimation, and classified as normal or depressed. The LV diameter was obtained at end-diastole, at the level of the papillary muscles, and was classified as normal or increased (>53 mm in women and >59 mm in men in the parasternal long-axis plane). The thickness of the interventricular septum and of the LV posterior wall was determined in diastole and was classed as either normal or increased (>10 mm). The anteroposterior diameter of the left atrium was measured at end-systole, in the parasternal long-axis plane, and was classed as either normal (<40 mm) or dilated (≥40 mm). The right ventricle (RV) was considered dilated if the RV basal diameter was >41 mm. Pericardial effusion was classed as either present or absent. The reference test for vegetation, valvular regurgitation and structural heart disease was TEE or TTE.

### 2.5. Statistical Analysis

Measures of central tendency and dispersion were calculated for the quantitative variables and absolute and relative frequencies for the qualitative ones. The normality of the variables was checked with the Kolmogorov–Smirnov test prior to the analysis to decide whether to use parametric or non-parametric tests. The qualitative and quantitative variables were analyzed using the chi-squared test and the Kruskal–Wallis test, respectively. In addition, 95% confidence intervals were obtained both for the mean values and for the proportions. To determine the accuracy of POCUS, a performance study of diagnostic tests was carried out, evaluating the sensitivity, specificity and positive and negative predictive values (with 95% confidence intervals) from the data obtained by POCUS and conventional echocardiography (reference test). The level of agreement between the results from the two diagnostic tests, in terms of the Kappa coefficient, was also determined. All statistical analyses were performed using the IBM^®^ SPSS v. 21 package.

## 3. Results

Between 1 February 2018 and 31 December 2020, 258 patients were included in this study. Characteristics of patients with bacteremia or candidemia as well as the contributions made by each participating hospital, and the POCUS outcomes, are detailed in Table 2. The mean age (SD) of the study population was 67.3 (14.7) years. Of these patients, 63.6% (164) were male. The mean hospital stay (SD) was 29.4 (25.4) days. The average time spent obtaining and interpreting the POCUS images was 12:56 (6:37) minutes.

Table 3 shows the proportion of cases of IE caused by different microorganisms. The main causes of positive blood cultures in IE patients were *Methicillin-sensitive staphylococcus aureus* (*n* = 15, 20.5%), *Enterococcus* spp. (*n* = 14, 19.2%), other *Streptococcu*s (*n* = 12, 16.4%) and *Coagulase-negative staphylococcus* (*n* = 11, 15%). A TEE was performed for 76 patients (29.5%). A diagnosis of definite IE was made for 64 patients (24.8%) and one of possible IE for another 9 (3.5%). Of the 88 cases of *Staphylococcus aureus* bacteremia (including MSSA and MRSA), a diagnosis of definite IE was made for 16 (18.2%) and one of possible IE for another 4 (4.5%).

Table 4 shows the characteristics of IE cases, including definite and possible, according to Duke criteria. Of the 73 cases of IE (definite or possible), 45 (61.6%) were acquired in the community and 28 (38.4%) were healthcare-related. Regarding the type of valve, 50 cases (68.5%) of IE concerned the native valve, 20 (27.4%) a prosthetic and 3 (4.1%) the pacemaker/ICD lead. By location, 33 cases (45.2%) were in the aortic valve, 30 (41.1%) were in the mitral valve, 5 (6.8%) were in the tricuspid valve, 2 (2.8%) were in the aortic and mitral valves and 3 (4.1%) were in the pacemaker/ICD. Cardiac surgery was performed in 21 (28.8%) patients with IE. In 16 (76.2%) cases, the surgery was scheduled and in 5 (23.8%) it was urgent. The indication for cardiac surgery was heart failure in 7 (33.3%) patients, uncontrolled infection in 11 (52.4%) and to prevent septic embolism in 3 (14.3%).

Table 5 shows the usefulness of POCUS for the detection of signs suggestive of IE in patients with bacteremia or candidemia. Using conventional echocardiography (TTE, TEE) as the reference method, the usefulness of POCUS in detecting valvular vegetation in patients with bacteremia or candidemia presented sensitivity, specificity and positive and negative predictive values of 77%, 94%, 82% and 92%, respectively. With respect to the detection of vegetation, the degree of agreement (concordance) between the internist (through POCUS) and the cardiologist (using conventional echocardiography) was 0.733. The sensitivity, specificity and positive and negative predictive values for the detection of vegetation on the aortic valve (Appendix A) were 61%, 98%, 81% and 95%, respectively, while the corresponding values for the mitral valve were 86%, 97%, 86% and 97%, respectively, and those for the tricuspid valve, 50%, 99%, 62% and 98%, respectively. The sensitivity of the diagnosis of valvular regurgitation (aortic, mitral or tricuspid) by POCUS compared to conventional echocardiography was ≥76%. The specificity was ≥90% for the other variables analyzed, except for mitral regurgitation (85%), aortic regurgitation (88%) (Appendix A) and tricuspid regurgitation (88%).

Table 6 shows the usefulness of POCUS for the detection of hepatomegaly, splenomegaly and septic embolisms, which are all common findings in IE (Figure 1). For this purpose, we analyzed a sample of 186 patients (72.1%) who underwent an abdominal imaging test (ultrasound, CT). Using this as a reference method, the usefulness of POCUS for the detection of hepatomegaly and splenomegaly presented sensitivity values of 92% and 92%, respectively, and specificity values of 96% and 98%, respectively. Septic embolisms were observed in seven patients (9.6%). However, the sensitivity in detecting hepatic and splenic infarction (Appendix A) was only 60% and 50%, respectively. It was not possible to properly assess the values for renal infarctions due to insufficient data.

## 4. Discussion

The study results obtained show that POCUS, as a complement to physical examination, when performed by an internist with formal training in echocardiography, in patients with bacteremia or candidemia, is a test that is valid and safe for the detection of signs suggestive of IE.

To our knowledge, this is the first multicenter study conducted to evaluate the usefulness of POCUS in patients with bacteremia or candidemia and clinical suspicion of IE.

Echocardiography (TTE, TEE) is the cornerstone imaging technique both in the diagnosis and management of IE, allowing for the detection of valvular vegetation and structural heart disease. However, its routine use is not always possible [7,8]. Technological improvements have led to the development of miniaturized handheld devices. Various studies have reported comparable results between handheld and standard echocardiograms concerning left ventricular systolic function, chamber size and valve disease [12,13,14,15,16]. POCUS provides greater sensitivity than physical examination in detecting significant valve disease [17,18]. In the present study, POCUS was used, in accordance with the recommendations of Narula et al. [19], as the fifth pillar of the physical examination at the hospital bedside.

In our study, the average time required to obtain and interpret the POCUS images was 12:56 min, similar to that reported by Jenkins et al. [20]. However, there were differences among the participating hospitals, which could limit the possible extrapolation of the results to the case of other users of this technology, with varying degrees of training and experience. Our results show that POCUS has an adequate sensitivity and high specificity for the detection of valvular vegetation, especially on the mitral valve. These are findings similar to those published recently by Bonzi M et al. [21] in a systematic review and meta-analysis to assess the diagnostic accuracy of TTE in patients with suspected IE of the native valves, describing a sensitivity and specificity of 0.71 (95% CI 0.56–0.82) and 0.80 (95% CI 0.85–0.92), respectively. They considered the use of a negative TTE as a single rule-out test in patients with a low pre-test probability of IE.

A literature review by Marbach et al. [16] on the role of POCUS reported a sensitivity of 82% and specificity of 89–99% for the detection of aortic regurgitation, corresponding values of 48–100% and 77–91% for mitral valve regurgitation and 65–89% and 89–98% for tricuspid valve regurgitation. These findings are of undoubted clinical importance, showing that the use of POCUS in patients with bacteremia or candidemia would enhance their stratification.

A recent study [22] showed that multi-organ POCUS provides relevant diagnostic information, complementing traditional physical examination including identification of heart valve lesions, and facilitates therapy adjustment in internal medicine wards, regardless of the cause of admission. In our study, we performed multi-organ POCUS, including lung ultrasound (LUS). It is well-known that valvular regurgitation (aortic, mitral) may lead to heart failure (HF). LUS can diagnose lung congestion by the detection of bilateral B-lines and therefore improve the diagnostic accuracy and management [23,24].

For the detection of significant heart murmurs, as well as conditions such as hepatomegaly and splenomegaly, which are commonly experienced by patients with IE, physical examination has well-known limitations, even when performed by expert clinicians [17,24]. Recently, López Zúñiga et al. [25] reported a sensitivity of 44% and specificity of 96.9% for the detection of hepatomegaly, as well as a sensitivity of 69.2% and specificity of 99.3% for the detection of splenomegaly, using POCUS in patients with suspected abdominal pathology. Our results showed a similar specificity and a slightly higher sensitivity. However, the usefulness of POCUS for the detection of hepatic, splenic and renal infarction in our study was limited.

Recently, Marbach JA et al. [26] described the usefulness of POCUS when considering infective endocarditis in patients with a notable regurgitation jet in the aortic and mitral valves, allowing for an approach to POCUS-guided clinical decision-making. In our study, we have found usefulness of POCUS in detecting signs suggestive of IE, including aortic and mitral regurgitation as well as hepatomegaly and splenomegaly, common findings in IE.

Our study has several limitations that should be commented on. First, the training and experience of the internists who performed POCUS was not homogeneous. To address this problem, it would be advisable to carry out a multicenter study including a standardized basic echocardiography training plan. Second, POCUS does not have pulsed Doppler, which limits its usefulness for evaluating diastolic dysfunction of the left ventricle (at present, therefore, intracardiac gradients cannot be quantified, nor can a quantitative assessment be made of valvular heart disease). However, vegetations and valvular regurgitation (characteristic findings in IE) can be detected by means of two-dimensional echocardiography and cooler Doppler, which are available with POCUS. Another limitation of the study was the low proportion of patients who underwent TEE (29.5%), which could have led to an infra-diagnosis of IE within the study population. Nevertheless, we considered a follow-up strategy in patients in which TEE was not performed, with TTE repetition at 7 days, as suggested by guidelines. In a previous study [27], of the total 241 patients with MRSA bacteremia, 114 (47.3%) were evaluated with TTE only, 56 (23.2%) with TEE only, and 71 (29.5%) TTE followed by TEE. Although TEE is a safe and generally well-tolerated procedure, as an invasive procedure it is not without risk and cost. Therefore, individual patient circumstances may justify different approaches.

Another point to be noted is that the diagnostic accuracy of conventional TTE in detecting vegetations in prosthetic valves is very limited. The European Society of Cardiology Guidelines has estimated the sensitivity of TTE for IE to be around 70% for native valves and 50% for prosthetic valves, with a specificity of approximately 90% [4]. In our study, we included 31 (12%) patients with prosthetic heart valve. To address this question, it would be appropriate to carry out a study with a more rigorous inclusion criteria, excluding patients with prosthetic valves and cardiac electrostimulation devices.

It is well-documented that the early use of echocardiography could affect the diagnostic accuracy in detecting valvular vegetation. TTE may fail to detect small vegetations. Therefore, in our study, we performed POCUS from the fifth day of the positive blood culture, and repeated it in 7 days, according to the current guidelines [4,9].

There is debate on whether TEE is necessary for evaluation of all *Staphylococcus aureus* bacteremia. Several risk factors predict IE in patients with indeterminate or positive TTE, community-acquired infection, intravenous drug use, high-risk cardiac conditions and prolonged *S. aureus* bacteremia > 72 h [11,27]. In our study, of the 88 cases of *S. aureus* bacteremia, a diagnosis of definite IE was made for 16 (18.2%) and one of possible IE for another 4 (4.5%), according to previous studies [10].

Recently, Sivak et al. [8] have proposed that an initial TTE with “strict negative criteria” (moderate or better ultrasound quality, normal anatomy, no valvular stenosis or sclerosis, less than mild valvular regurgitation, less than moderate pericardial effusion, absence of pacemaker/defibrillator leads, absence of prosthetic valves and absence of typical or suggestive signs of IE) predicts both a low probability of requesting a follow-up study and of a definitive diagnosis of IE. The findings of our study suggest that POCUS could be an adequate test in patients with bacteremia or candidemia and low clinical probability of IE, due to the high negative predictive value for the detection of valvular vegetation, pericardial effusion, hepatomegaly and splenomegaly.

Among its advantages, POCUS can be performed by the attending clinician, at the hospital bedside, in real time, providing a qualitative or semi-quantitative assessment of the heart anatomy and function in response to specific clinical questions, usually providing dichotomous data (e.g., the presence or absence of significant valvular heart disease). This information can then be integrated into the decision-making process, helping optimize clinical–therapeutic algorithms, avoiding diagnostic delay, improving the prognosis and, probably, strengthening the doctor–patient relationship, thus regaining “the art of medicine” [12,13,14,15,16,17,18,19]. Nevertheless, and despite these apparent advantages, POCUS requires a regulated training process and adequate experience. It cannot replace conventional echocardiography or clinical judgment in establishing a firm diagnosis of IE, so POCUS cannot be used to rule-in or rule-out IE. POCUS should not be considered as a diagnostic test, but rather as an extension of the physical examination, the same way as the stethoscope, introduced by Laennec.

## 5. Conclusions

The proposed use of POCUS in clinical practice could be an extension of physical examinations at the hospital bedside in patients with bacteremia or candidemia, helping to identify signs suggestive of IE.

## Figures and Tables

**Figure 1 jcm-11-03636-f001:**
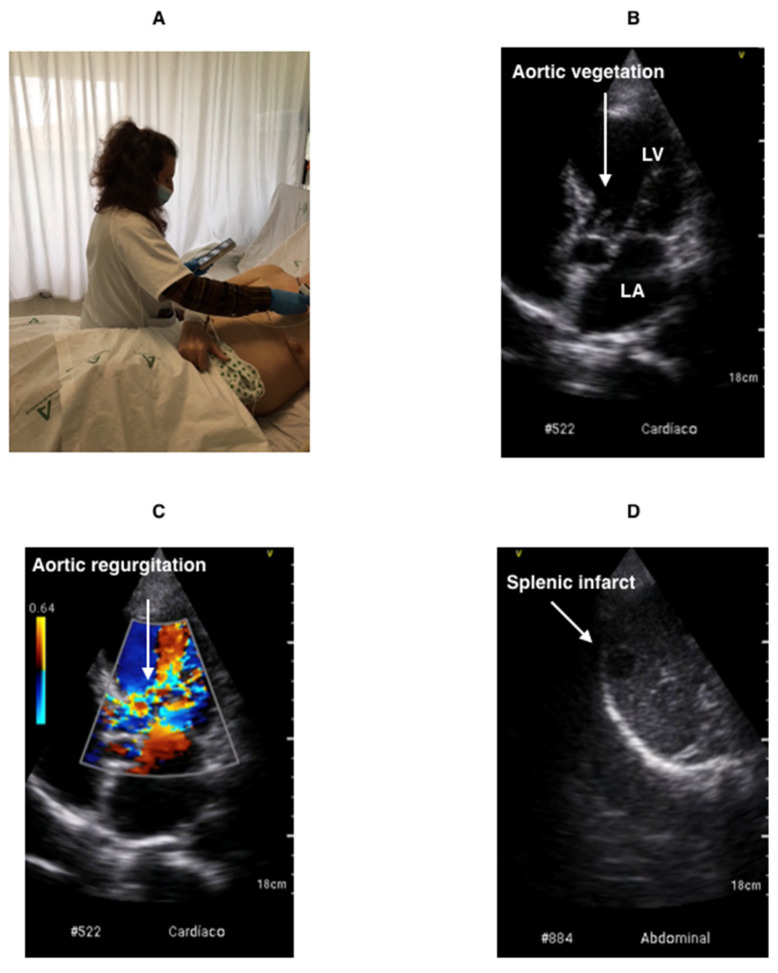
Demonstration of the POCUS technique. (**A**). POCUS at the patient’s hospital bedside. (**B**). Vegetation on aortic valve. (**C**). Severe aortic regurgitation. (**D**). Splenic infarction. LA: left atrium. LV: left ventricle.

**Table 1 jcm-11-03636-t001:** Microbiology.

*Staphylococcus aureus*
*Viridans streptococcus*: *S. mitis*, *S. sanguis*, *S. mutans*, *S. milleri*, *S. salivarius*
*Streptococcus gallolyticus* (formerly, *S. bovis*)
HACEK group: *Haemophilus aphrophilus* (subsequently called *Aggregatibacter aphrophilus*), *Actinobacillus actinomycetemcomitans* (subsequently called *Aggregatibacter actinomycetemcomitans*), *Cardiobacterium hominis*, *Eikenella corrodens*, *Kingella kingae*.
*Enterococcus* spp.
Microorganisms usually considered epithelial contaminants, isolation in three or four blood culture flasks: *Coagulase-negative staphylococcus*, *Corynebacterium* spp., *Cutybacterium acnes*, *Bacillus* spp.
*Candida* spp.
Others: *Pseudomonas aeruginosa*, *Acinetobacter* spp., *Coxiella burnetti* (IgG phase I > 1:800), *Brucella* spp., *Bartonella* spp., *Chlamydia psittaci*, *Legionella* spp., *Mycoplasma* spp., *Tropheryma whippelii*, *Lactobacillus* spp., *Gordonia bronchialis*, *Erysipelothrix rhusiopathiae*, *Neisseria elongata*, *Moraxella catarrhalis*, *Veillonella* spp., *Listeria monocytogenes*, *Campylobacter fetus*, *Francisella tularensis*, *Catabacter hongkongensi*

**Table 2 jcm-11-03636-t002:** Characteristics of patients with bacteremia or candidemia.

Patients *n*: 258	H.U.T.	H.U.J	H.U.To.	H.U.MS	H.U.VB	H.U.V.N.	*p*
Patients recruited, *n* (%)	101 (39.2%)	54 (20.9%)	45 (17.4%)	38 (14.7%)	12 (4.7%)	8 (3.8%)	0.001 *
Age, years (mean)	65.1	67.8	71.1	67.7	66.7	70.8	0.776 **
Gender (female), *n* (%)	38 (37.6%)	20 (37%)	12 (26.6%)	17 (44.7%)	4 (33.3%)	3 (37.5%)	0.683 *
Number of POCUS examinations/year performed by involved operators	500	450	220	250	350	150	<0.001 *
Clinical manifestations, *n* (%)							
Fever	93 (92.1%)	53 (98.14%)	45 (100%)	38 (100%)	12 (100%)	8 (100%)	0.074 *
Duration of the fever, days (mean)	2.8	1.8	1.9	1.7	2.6	2.5	0.182 **
Shivering	64 (63.3%)	36 (66.6%)	32 (71.1%)	14 (36.8%)	7 (58.3%)	6 (75%)	0.024 *
Anorexia	65 (64.3%)	29 (53.7%)	31 (68.8%)	15 (39.5%)	4 (%)	4 (50%)	0.026 *
Weight loss	32 (31.7%)	20 (37%)	18 (40%)	5 (13.1%)	3 (33.3%)	1 (12.5%)	0.080 *
Dyspnoea	31 (30.7%)	22 (40.7%)	17 (33.7%)	11 (28.9%)	6 (50%)	3 (37.5%)	0.611 *
Myalgia	36 (35.6%)	17 (31.5%)	11 (22.4%)	3 (7.9%)	5 (41.6%)	2 (25%)	0.034 *
Night sweats	34 (33.6%)	16 (29.6%)	14 (31.1%)	4 (10.5%)	3 (25%)	3 (37.5%)	0.163 *
Heart murmur	33 (32.6%)	13 (24%)	16 (35.5%)	10 (26.3%)	5 (41.6%)	3 (37.5%)	0.705 *
Risk factors, *n* (%)							
Prosthetic heart valve	3 (2.9%)	11 (20%)	10 (22%)	3 (7.9%)	3 (25%)	1 (12.5%)	0.003 *
Congenital heart disease	0	3 (5.5%)	1 (2.2%)	0	0	0	0.140 *
Permanent pacemaker	8 (7.9%)	3 (5.5%)	9 (20%)	1 (2.6%)	2 (16.6%)	0	0.048 *
ICD	2 (1.9%)	2 (3.7%)	1 (2.2%)	2 (5.2%)	0	0	0.857 *
Charlson Index (mean)	3.5	4.2	4.7	2.9	4.1	4.1	0.014 **
Deficient oral hygiene	49 (48.5%)	11 (20.3%)	23 (51.1%)	4 (10.5%)	6 (50%)	1 (12.5%)	0.001 *
Dental procedures	2 (1.9%)	2 (3.7%)	3 (6.6%)	2 (5.3%)	0	0	0.682 *
Endoscopy	14 (13.8%)	4 (7.4%)	4 (8.8%)	4 (10.5%)	1 (8.3%)	1 (12.5%)	0.864 *
Surgery	24 (23.7%)	6 (11.1%)	2 (4.4%)	5 (13.2%)	1 (8.3%)	1 (12.5%)	0.049 *
Immunosuppression	24 (23.8%)	11 (20.4%)	9 (20%)	15 (39.5%)	3 (25%)	2 (25%)	0.350 *
Diabetes mellitus	39 (38.6%)	17 (31.5%)	18 (40%)	12 (31.6%)	4 (33.3%)	5 (62.5%)	0.582 *
Chronic kidney disease	29 (28.7%)	14 (25.9%)	5 (11.1%)	4 (10.5%)	4 (33.3%)	2 (25%)	0.078 *
Microbiology, *n* (%)							
MSSA	25 (24.7%)	15 (27.7%)	12 (26.6%)	7 (18.4%)	3 (25%)	2 (25%)	0.948 *
MRSA	5 (4.9%)	2 (3.7%)	5 (11.1%)	9 (23.7%)	2 (16.6%)	1 (12.5%)	0.012 *
CNS	16 (5.9%)	8 (14.8%)	5 (11.1%)	4 (10.5%)	0	0	0.512 *
Viridans streptococci	6 (5.9%)	1 (1.8%)	4 (8.8%)	4 (10.5%)	3 (25%)	2 (25%)	0.036 *
Other streptococci	14 (13.8%)	8 (14.8%)	6 (13.3%)	2 (5.3%)	4 (33.3%)	0	0.172 *
*Enterococci* spp.	14 (13.8%)	14 (25.9%)	9 (20%)	2 (5.3%)	0	0	0.031 *
*Candida* spp.	19 (18.8%)	3 (5.5%)	1 (2.2%)	8 (21.1%)	0	2 (25%)	0.009 *
Other microorganisms	2 (1.9%)	3 (5.5%)	3 (6.6%)	2 (5.3%)	0	1 (12.5%)	0.534 *
Duration of POCUS examination and interpretation, minutes (mean)	9:43	08:26	23:39	15:36	15:02	11:03	0.001 **

* Chi-squared; ** Kruskal-Wallis; ICD: Implantable Cardioverter–Defibrillator. MSSA: Methicillin-sensitive staphylococcus aureus. MRSA: Methicillin-resistant Staphylococcus aureus. CNS: Coagulase-negative staphylococcus. H.U.T.: Torrecárdenas University Hospital, Almería. H.U.J: Jaén University Hospital. H.U.To: Toledo University Hospital. H.U.MS: Marina Salud University Hospital, Denia. H.U.VB: Vega Baja University Hospital, Orihuela. H.U.V.N.: Virgen de las Nieves University Hospital, Granada.

**Table 3 jcm-11-03636-t003:** Proportion of cases of IE caused by different microorganisms.

	Definite (*n*)	Possible (*n*)	Rejected (*n*)
MSSA	13	2	49
MRSA	3	2	19
CNS	10	1	22
Viridans streptococcus	8	1	11
Other streptococcus	11	1	22
*Enterococcus* spp.	13	1	25
*Candida* spp.	0	1	32
Other microorganisms	6	9	5
Total, *n* (%)	64 (24.8%)	9 (3.5%)	185 (71.7%)

MSSA: Methicillin-sensitive staphylococcus aureus. MRSA: Methicillin-resistant Staphylococcus aureus. CNS: Coagulase-negative staphylococcus.

**Table 4 jcm-11-03636-t004:** Characteristics of IE cases (Definite and Possible).

Patients (*n*)	73
Location, *n* (%)	
Aortic	33 (45.2%)
Mitral	30 (41.1%)
Tricuspid	5 (6.8%)
Aortic + mitral	2 (2.7%)
Pacemaker lead/ICD	3 (4.1%)
Type of valve, *n* (%)	
Native	50 (68.5%)
Prosthetic	20 (27.4%)
Pacemaker lead/ICD	3 (4.1%)
Acquisition, *n* (%)	
Community	45 (61.6%)
Healthcare-related	28 (38.4%)
Heart surgery, *n* (%)	
Scheduled	16 (76.2%)
Urgent	5 (23.8%)
Indication for heart surgery, *n* (%)	
Heart failure	7 (33.3%)
Uncontrolled infection	11 (5.,4%)
Prevent septic embolism	3 (14.3%)

**Table 5 jcm-11-03636-t005:** Usefulness of POCUS for the detection of signs suggestive of IE in patients with bacteremia or candidemia.

	TP	FP	FN	TN	Sensitivity (95% CI)	Specificity (95% CI)	PPV (95% CI)	NPV (95% CI)	Concordance (Kappa)
Valvular vegetation	52	11	15	180	0.77	0.94	0.82	0.92	0.733
					(0.67–0.87)	(0.90–0.97)	(0.61–0.92)	(0.88–0.96)	
Aortic valve vegetation	17	4	11	226	0.61	0.98	0.81	0.95	0.662
					(0.43–0.79)	(0.97–0.99)	(0.61–0.99)	(0.92–0.98)	
Mitral valve vegetation	30	5	5	218	0.86	0.97	0.86	0.97	0.835
					(0.74–0.97)	(0.96–0.99)	(0.72–0.98)	(0.95–0.99)	
Tricuspid valve vegetation	5	3	5	245	0.5	0.99	0.62	0.98	0.540
					(0.19–0.81)	(0.97–1)	(0.22–0.99)	(0.96–0.99)	
Aortic regurgitation *	67	20	21	150	0.76	0.88	0.77	0.87	0.645
					(0.67–0.85)	(0.83–0.93)	(0.67–0.86)	(0.82–0.92)	
Mitral regurgitation *	91	20	28	119	0.76	0.85	0.82	0.81	0.624
					(0.69–0.84)	(0.79–0.91)	(0.74–0.89)	(0.74–0.87)	
Tricuspid regurgitation *	81	18	23	136	0.77	0.88	0.82	0.85	0.667
					(0.69–0.85)	(0.83–0.93)	(0.73–0.89)	(0.79–0.91)	
LV systolic dysfunction	22	5	5	226	0.81	0.98	0.81	0.98	0.793
					(0.66–0.96)	(0.95–0.99)	(0.65–0.98)	(0.96–0.99)	
LV dilatation	15	3	4	236	0.79	0.98	0.83	0.98	0.796
					(0.60–0.97)	(0.97–1)	(0.63–0.99)	(0.96–0.99)	
LA dilatation	92	14	18	134	0.84	0.90	0.86	0.88	0.745
					(0.76–0.90)	(0.85–0.95)	(0.80–0.94)	(0.83–0.94)	
RA dilatation	37	11	10	200	0.78	0.95	0.77	0.95	0.729
					(0.67–0.90)	(0.91–0.97)	(0.64–0.90)	(0.92–0.98)	
RV dilatation	17	8	4	229	0.81	0.96	0.68	0.98	0.714
					(0.64–0.97)	(0.97–0.99)	(0.47–0.88)	(0.96–0.99)	
Pericardial effusion	14	9	2	233	0.87	0.96	0.61	0.99	0.696
					(0.71–1)	(0.94–0.99)	(0.39–0.83)	(0.98–0.99)	

TP: True positive. FP: False positive. FN: False negative. TN: True negative. PPV: Positive predictive value. NPV: Negative predictive value. LV: Left ventricle. AI: Left atrium. RA: Right atrium. RV: Right ventricle. * More than mild valvular regurgitation

**Table 6 jcm-11-03636-t006:** Usefulness of POCUS for the detection of hepatomegaly, splenomegaly and septic embolisms in patients with bacteremia or candidemia.

	TP	FP	FN	TN	Sensitivity (95% CI)	Specificity (95% CI)	PPV (95% CI)	NPV (95% CI)	Concordance (Kappa)
Hepatomegaly	12	7	1	166	0.92	0.96	0.63	0.99	0.727
					(0.77–1)	(0.93–0.99)	(0.39–0.87)	(0.98–1)	
Splenomegaly	11	3	1	171	0.92	0.98	0.78	0.99	0.935
					(0.76–1)	(0.96–1)	(0.53–1)	(0.98–1)	
Hepatic infarction	3	1	2	180	0.6	0.99	0.75	0.99	0.659
					(0.17–1)	(0.98–1)	(0.20–0.99)	(0.97–1)	
Splenic infarction	4	0	4	178	0.5	1	1	0.98	0.657
					(0.15–0.84)	(1–1)	(0.87–1)	(0.95–1)	
Renal infarction	0	1	5	180	-	0.99	-	0.97	-
						(0.98–1)		(0.94–0.99)	

TP: True positive. FP: False positive. FN: False negative. TN: True negative. PPV: Positive predictive value. NPV: Negative predictive value.

## Data Availability

The authors confirm that the data supporting the findings of this study are available from the corresponding author, upon reasonable request.

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
