# Peer review of "Point-of-Care Ultrasound (POCUS) as an Extension of the Physical Examination in Patients with Bacteremia or Candidemia"

_jcm, 2022, doi:10.3390/jcm11133636_

Round 1
Reviewer 1 Report
This paper was an interesting observational study in 6 hospitals in Spain, aiming to compare point-of-care ultrasound (POCUS) to conventional echo for the detection of valvular vegetation and abnormalities. Here are some concerns and suggestions to improve the manuscript:
1) Abstract: I suggest for the Objective section, consider "determine" instead of "analyse". You discuss septic embolism in the objectives but not in the results. Minor typo - Patients: please fix "adult patints". It was not clear with valvular regurgitation, the methods discuss grading regurgitation as mild, moderate, or severe. It is unclear if you are reporting the sensitivity/specificity of POCUS to detect the presence of valvular regurgitation or the severity of valvular regurgitation.
2) Introduction: could be improved by separating into 3 or 4 short paragraphs
3) Methods: under patient selection, the list of positive blood culture organisms would be better suited for a table. I suggest including % excluded for no TTE or TEE and very poor windows here in the methods, and remove from results. The appendix forms do not match in terms of ultrasound window quality - the appendix uses "good or bad" and "very poor" is used in the methods. This should match, unless you weren't excluding patients with "bad" windows.
4) Methods 2.3 - It would be important to know how often the POCUS internists were using POCUS, given the high variability in time it took to do and interpret the studies. How did you ensure blinding?
5) Figure 1 - formatting could be improved, and I suggest arrows to point out the findings.
6) Table 1 - I suggest using % female under Sex, as the female/male numbers are less clear. Was it one internist at each hospital performing all the POCUS studies? Should there be a range of experience with POCUS for each hospital? For the duration of POCUS examination, the text suggests this number includes interpretation. I would make sure this is clear
7) Table 2 - same question as in the abstract. It is unclear if you are reporting the sensitivity/specificity of POCUS to detect the presence of valvular regurgitation or the severity of valvular regurgitation.
8) Results - the last paragraph discussing that 25.6% of POCUS exams aided in the diagnosis and or treatment. I would introduce this objective in the methods and more clearly define what you mean by how the POCUS exam changed the clinical course.
9) Conclusions - minor typo in line 301, fix "showed that". These could be strengthened as the methods/results are more clearly presented.
Author Response
Dear Reviewer
Firstly, we would like to thank you for your effort to review our manuscript. Next, we are going to explain, point by point, the details of the revisions to the manuscript and our responses to your referees´comments:
1.) Abstract. We change “determine” for “analyse”. We have fixed “patints” for “patients”. The expression “more than mild regurgitation” has been introduced.
2.) Introduction. 4 short paragraphs have been introduced, as suggested.
3.) Methods. Table with the list of positive blood culture organisms has been introduced. We have included “% excluded for no TTE or TEE and very poor windows” in methods and removed from results.
4.) Methods 2.3. We have introduced how often POCUS is used by internists.
5.) Figure 1. Formatting has been improved including arrows as suggested.
6.) Table 2. % female under Sex has been modified. We have introduced the number of examinations/year performed by involved operators.
7.) Table 5. The concept “more than mild valvular regurgitation” has been introduced.
8.) Results. Paragraph “25.6% of POCUS exams aided in the diagnosis and or treatment …” has been removed to avoid misunderstanding.
9.) Conclusions. “Showed that” has been modified.
Thanks to your suggestions we consider that our manuscript has improved considerably.
Yours faithfully.

Reviewer 2 Report
The paper by Lopez Palmero et al addresses the use of POCUS in the diagnostic workup of Infective Endocarditis (IE). The use of POCUS is rapidly growing and its benefits in terms of increasing diagnostic accuracy of physical examination and speeding up diagnosis at the bedside have been widely demonstrated in the literature, thus the topic of the paper could be of interest. Nevertheless, I have several relevant concerns about the methodology and the conclusions making in my opinion the paper very weak as a whole.
- As recognized by the Authors, the gold standard test (TE) was performed in a minority of patients: this is in my opinion a major limitation markedly affecting the results and their interpretation. It should have been considered -at least- a follow-up strategy in patients in which TE was not performed, with TT repetition at 7 days (as suggested by guidelines). Moreover, exclusion of patients with poor ultrasound window could have overestimate sensitivity.
-Inclusion of prosthetic valve represents a real hazard. Patients with prosthetic valves and devices have to be excluded.
-The meaning of the “early dectection” can be misleading: “early” can be intended both as a test that is performed for screening purposes before another test (but index test for screening require high sensitivity which is not the case) or can refer to ‘early in the endocarditis course’. The latter could be tricky as ‘early‘ vegetation can be smaller and more difficult to detect, even for TT and TE.
-Methods section, pag 2 line 112: rather than the years of experience it should be specified the number of examinations/year performed by involved operators. Moreover, it should be stated if they were expert only in POCUS or they perform complete TT in their routine activity.
-Results, pag 5 line 198. What do the Authors mean exactly with “because conventional echocardiography was not available”? Non availability of echocardiography is not acceptable in non-limited resource setting.
-The conclusion that POCUS ‘ is a safe, valid test for the early detection of infective endocarditis” is not pertinent at all: sensitivity of POCUS is not satisfying for this purpose. Should we accept a test with a sensitivity of 75% to rule-out a high mortality condition? This is sensitivity for vegetation recognition, but is not clear how the Authors consider the ‘positive’ or ‘negative’ POCUS, being the items considered separately and compared to TTE and TE, not to the final diagnosis of endocarditis: Authors should explain the possible use and added value of POCUS in the diagnosis of endocarditis: in which case a ‘positive POCUS’ or a ‘negative POCUS’ should change/fasten the diagnostic workup of the patients? Consider that ‘Strict negative’ criteria are required to rule out IE (Marcos-Garces V et al, Am J Cardiol 2022 1;162:156-162). Considering the results, I think the Authors should come to opposite conclusions: POCUS cannot be used to rule-in or rule-out IE.
Minor comments
-In the introduction line 56 pag 2 , describing sensitivity and specificity of TT for IE diagnosis the Authors cite a metanalysis and a small study (CIT 7-8). It should be more appropriate to add another metanalysis (Diagnostic accuracy of transthoracic echocardiography to identify native valve infective endocarditis: a systematic review and meta-analysis. PMID: 29546685 DOI: 10.1007/s11739-018-1831-0) that focus on native valves. The metanalysis by Bai at al comprise bot native and prosthetic valves, the latter are not matter of discussion for a POCUS (maybe also for a TT).
-In the Introduction pag 2 line 66-67: risks of TE are too overemphasized: these are very rare complications. Cost, patients discomfort, hospitalization prolongation, lack of prompt TE availability are more often the main issues facing with TE execution. please modify the statement
-In the Introduction pag 2 line 67-70: “TEE is required in the following clinical scenarios: the presence of a prosthetic valve or a pacemaker, TTE negative or technically inadequate with a strong clinical suspicion of IE, positive TTE and suspicion of intracardiac complications (such as an abscess), and prior to cardiac surgery” : it should be more appropriate to refer to endocarditis guidelines (doi.org/10.1093/eurheartj/ehv319) rather than on single study /metanalysis. Moreover, there isn’t unanimous consensus about reserving TE only in positive TT with suspicion of complication (most clinicians perform TE in every positive TT as TT examination is not enough sensitive to detect signs of complication).
-table 4: it should be specified in the legend the type of regurgitation: the sensitivity /specificity refers to any type of regurgitation? Or a regurgitation “more than trivial” (the latter is the most used definition in endocarditis clinical trials).
Pag 181: right ventricle diameter: to which guideline this cut-off refers to? Normality cut-off for the diastolic right ventricle diameter is 41mm (Lang et al, JASE 2015)
Author Response
Dear Reviewer
Firstly, we would like to thank you for your effort to review our manuscript. Next, we are going to explain, point by point, the details of the revisions to the manuscript and our responses to your referees´comments:
1.) Introduction. We have added another metanalysis (doi: 10.1007/s/11739-018-1831-0) as suggested. Risks of TEE have been modified as suggested. TEE recommendations have been referred to endocarditis guidelines.
2.) Methods. We have introduced a follow-up strategy in patients in which TEE was not performed, with TTE repetition at 7 days as suggested by guidelines. Follow-up TTE was performed more frequently in patients with persistent bacteremia or candidemia (positive blood cultures 48 hours after starting appropriate antimicrobial treatment) and, specifically, when blood cultures were positive for Enterococcus spp, Staphylococcus aureus or Streptococcus spp. The expression “because conventional echocardiography was not available” has been modified. “Cutt-off for the diastolic right ventricle diameter” has been modified as suggested
3.) Table 2. We have specified the number of examinations/year performed by involved operators as suggested.
4.) Table 4. The legend “more than mild regurgitation” has been introduced as suggested.
4.) Discussion. Inclusion of prosthetic valve represents a real hazard. This issue has been described as a major limitation of our study. To address this question it would be appropriate to carry out a study with a more rigorous inclusion criteria, excluding patients with prosthetic valves and cardiac electroestimulation device.
5.) Conclusion. The expression “early detection” has been removed to avoid misunderstanding. “POCUS is a safe, valid test for the early detection of IE” has been modified. We have highlighted that POCUS cannot be used to rule-in or rule-out IE.
Thanks to your suggestions we consider that our manuscript has improved considerably.
Yours faithfully.

Round 2
Reviewer 1 Report
The article has been substantially improved. It looks like some of the comments have not been addressed, such as why the discrepancy between the echo forms and methods ("bad" vs. "poor" quality images).
Author Response
Reviewer 1. Round 2
Dear Reviewer
Firstly, we would like to thank you for your effort to review our manuscript.
Next, we are going to explain, point by point, the details of the revisions to the manuscript and our responses to your referees´comments:
1.) The concepts “bad”, “poor” and “very poor” have been included with the meaning of “bad ultrasound window”, as suggested, to avoid misunderstanding.
Thanks to your suggestions we consider that our manuscript has improved considerably.
Yours faithfully.
Reviewer 2 Report
The Authors addressed only partially the comments/criticisms presented in the first Review.
In particular, they did not change significantly the message conveyed by the paper: the study does not address the diagnostic accuracy of POCUS for IE, bu rather the concordance between POCUS and TT/TEE echocardiographic findings. This is a major difference and I think that the title and the text should be changed accordingly. The Authors should explain in which way POCUS could change/fasten the diagnostic approach of IE.
Moreover, the Authors added that 7 days follow-up was performed but did not show any data at this regard (In how many patients POCUS findings changed in the follow up? Which finding did change?How change the management of the patient accordingly?)
Author Response
Reviewer 2. Round 2
Dear Reviewer
Firstly, we would like to thank you for your effort to review our manuscript.
Next, we are going to explain, point by point, the details of the revisions to the manuscript and our responses to your referees´comments:
- The study does not address the diagnostic accuracy of POCUS for IE. According to the reviewer´s suggestion, we have changed the title and the text. Effectively, the main aim of the present study was not to address the diagnostic accuracy of POCUS for IE but rather to determine the usefulness of multi-organ POCUS, including cardiac, lung and abdominal ultrasound as an extension of the physical examination in patients with bacteremia or candidemia to detect signs suggestive of IE (e.g. valvular regurgitation, hepatomegaly, splenomegaly, liver infarcts, splenic infarcts) as well as to diagnose lung congestion by the detection of bilateral B-lines and pleural effusion. It is well known that valvular regurgitation (aortic, mitral) may lead to heart failure. In conclusion, POCUS should not be considered as a diagnostic test, but rather as an extension of the physical examination at bedside patient the same way the stethoscope was introduced by Laennec to improve physical examinations.
- The Authors should explain in which way POCUS could change/fasten the diagnostic approach of IE. Recently, Marbach JA et al (doi: 10.1016/j.chest.2020.07.021) described the usefulness of POCUS to consider infective endocarditis in those patients with a notable regurgitation jet in aortic and mitral valves, allowing an approach to POCUS-guided clinical decision-making. In our study, we have found usefulness of POCUS to detect signs suggestive of IE, including aortic and mitral regurgitation (more than mild) as well as hepatomegaly and splenomegaly, common findings in IE.
- The Authors added that 7 days follow-up was performed but did not show any data at this regard (In how many patients POCUS findings changed in the follow up? Which finding did change? How change the management of the patient accordingly?). In the clinical practice, we considered a follow-up strategy in patients in which TEE was not performed, with TTE repetition at 7 days, as suggested by guidelines. However, this variable was not defined in our study, so unfortunately we can not answer this issue.
Thanks to your suggestions we consider that our manuscript has improved considerably.
Yours faithfully.
Round 3
Reviewer 2 Report
The Authors did not addressed sufficiently the criticisms presented because they did not collect accordingly the data and/or they defined the aim of the study differently from what I think should have done, as they clarify in the second review. Anyway I think the paper has been improved and the amount of collected data deserves the publication.